# Acceleration-Based Estimation of Vertical Ground Reaction Forces during Running: A Comparison of Methods across Running Speeds, Surfaces, and Foot Strike Patterns

**DOI:** 10.3390/s23218719

**Published:** 2023-10-25

**Authors:** Dovin Kiernan, Brandon Ng, David A. Hawkins

**Affiliations:** 1Biomedical Engineering Graduate Group, University of California, Davis, Davis, CA 95616, USA; 2Department of Biomedical Engineering, University of California, Davis, Davis, CA 95616, USA; 3Department of Neurobiology, Physiology, & Behavior, University of California, Davis, Davis, CA 95616, USA

**Keywords:** wearables inertial measurement units (IMUs), in-field kinetics, over-ground gait biomechanics, machine learning

## Abstract

Twenty-seven methods of estimating vertical ground reaction force first peak, loading rate, second peak, average, and/or time series from a single wearable accelerometer worn on the shank or approximate center of mass during running were compared. Force estimation errors were quantified for 74 participants across different running surfaces, speeds, and foot strike angles and biases, repeatability coefficients, and limits of agreement were modeled with linear mixed effects to quantify the accuracy, reliability, and precision. Several methods accurately and reliably estimated the first peak and loading rate, however, none could do so precisely (the limits of agreement exceeded ±65% of target values). Thus, we do not recommend first peak or loading rate estimation from accelerometers with the methods currently available. In contrast, the second peak, average, and time series could all be estimated accurately, reliably, and precisely with several different methods. Of these, we recommend the ‘Pogson’ methods due to their accuracy, reliability, and precision as well as their stability across surfaces, speeds, and foot strike angles.

## 1. Introduction

Ground reaction forces (GRFs) are external reaction forces created with equal magnitude and opposite sense to the force that the foot applies to the ground with each step. Quantifying GRFs is fundamental to running biomechanics research because: (1) per Newton’s second law, GRFs dictate center of mass (COM) acceleration and can therefore be used to study whole-body motion; (2) this whole-body motion both causes, and is caused by, muscle activity and thus GRF provides insight into that activity [1]; (3) in combination with this muscle activity, GRF contributes to the internal loads experienced by structures within the body (e.g., bone, ligament, tendon, cartilage), leading to its frequent (though much contested) investigation as a risk-factor for injury [2,3,4,5,6,7,8,9,10]; (4) the magnitude and sense of GRF is used to assess running performance [11,12,13,14]; finally, (5) because GRFs are critical to inverse dynamics calculations, allowing for the estimation of joint forces and moments and more advanced analysis of behavior. Thus, accurate quantification of GRF during running is an important goal.

Accurate quantification of GRF is relatively easy with a force plate or instrumented treadmill, however, this equipment is generally ‘captive’ to lab environments that may not represent the actual conditions that runners experience. To increase ecological validity, previous research has attempted to replicate field conditions within the lab [15,16,17,18,19]. Even with such attempts, however, spatial constraints may cause participants to alter their gait on a treadmill or short running track [20], and temporal constraints still limit the duration and volume of data collection. Thus, lab measured GRFs may not accurately represent the millions of GRFs that occur over many long bouts of running in the field, limiting our understanding of GRF and its relation to other variables.

Some of these constraints have been overcome through the advancement of force sensing insoles [21,22,23,24,25,26], wearable load cells [27,28,29], and instrumented shoes [30,31,32]. However, this equipment still suffers from issues with durability, comfort, changing the mechanical properties of a shoe (i.e., making it more rigid) and interfering with the foot–ground (or foot–shoe) interface. Consequently, biomechanists remain largely reliant on ‘captive’ technology to measure GRF, decreasing the ecological validity and volume of available data.

Accelerometers offer a promising alternative to overcome this reliance on ‘captive’ technology. These small, low-cost wearable devices may allow the capture of greater volumes of more ecologically valid data than traditional ‘captive’ equipment. Sets of multiple wearables can estimate GRFs during walking [33,34,35,36] and running [37,38,39] and several methods have been proposed that truly capitalize on the advantages of accelerometers (minimizing system complexity, preparation time, participant discomfort, and costs) by estimating GRFs with a single accelerometer. These single-accelerometer methods often place the accelerometer on the shank or locations intended to approximate whole body COM such as the hip, lower back, or upper back (other locations such as the wrist have also been investigated but show poor correlations with GRFs [40,41]).

Researchers attempting to estimate GRFs from shank accelerations build on observations that shank acceleration and GRF signals are closely related [42,43,44] and therefore should allow for the estimation of one from the other [45,46,47]. Furthermore, some argue that forces applied at the ground are damped as they travel up the body and thus, measuring accelerations at the shank better reflects GRFs than more proximal mounting locations [48]. Researchers also point to similarities in the timing of peaks in the tibial acceleration and GRF signals [49] (cf. [50]) and argue for a mechanical coupling of these peaks [51].

Relations between GRF and accelerometers mounted on the approximate COM have also been explored [40,52]. Use of COM locations is based on Newton’s second law that states that whole body acceleration is inversely proportional to the mass of a body and proportional to the net forces acting on that body [53,54,55,56]. Given a constant mass, if forces other than GRF are relatively small (e.g., air resistance), then COM acceleration is a function of gravitational force and GRF. A limitation of this ‘COM’ approach is the assumption that a single accelerometer with a static position can capture whole body COM acceleration even though the COM location can move during running (due to limb movements and changes in posture). Despite this movement, previous research supports this assumption and demonstrates that a sacrum-mounted accelerometer captures whole body COM acceleration during running fairly well [57,58,59,60,61].

Based on these arguments, there have been many attempts to estimate GRF from acceleration at these two mounting locations [62]. However, only one study has conducted a head-to-head comparison of methods and that study only compared two of the many methods available [63]. Thus, there are no comprehensive recommendations to guide users on which method to use for a given application. To overcome this gap in the literature, methods to estimate GRF from a single accelerometer were replicated and compared. Methods were required to be non-participant or -trial specific (cf. [64]), non-proprietary (cf. [50,65,66]), report promising results (cf. [67]), and be capable of providing stance-by-stance estimates of at least one feature of the vertical GRF (first peak, loading rate, second peak, average, or time series) (cf. [56]) using only easy-to-measure anthropometrics and/or features of an acceleration input signal (cf. [16,56,68,69]) from a single sensor on the shank or COM (cf. [37,38,39,70,71,72]). In total, 27 methods derived from 13 publications met these criteria The 13 original publications are described in Table 1 while the methods derived from those publications and the vertical GRF feature they are capable of estimating are described in Table 2.

To evaluate which of these 27 methods provides the most accurate, reliable, and precise estimate of the vertical GRF first peak, loading rate, second peak, average, and/or time series, errors were calculated relative to a gold-standard force plate. For each method, errors were compared across a range of speeds, foot strike angles, and running surfaces to explore whether the method’s performance varied across these conditions. The results demonstrate the best method to estimate vertical GRF parameters from a single accelerometer under given surface, foot strike, and running speed conditions. Code to automatically execute each of the methods on stance-segmented accelerometer data is provided at https://github.com/DovinKiernan/MTFBWY_running_vGRF_from_a, accessed on 12 September 2023.

## 2. Methods

Data from this study were first reported in a separate analysis [15], but methods are repeated here for convenience.

### 2.1. IMU Calibration

Adapting methods from Coolbaugh et al. [81], tri-axial IMUs (ProMove MINI, Inertia Technology, Enschede, The Netherlands; ±16 g primary, ±100 g secondary, ±34.91 rad/s, 1000 Hz; see https://inertia-technology.com/wp-content/uploads/2022/02/ProMove-mini-datasheet.pdf, (accessed on 11 Oct 2023) for further details on device specifications and operation) were calibrated with a centrifuge (ClearPath MCVC, Teknic, Victor, NY, USA) and custom 3D printed jigs (SOLIDWORKS 2019, Dassault Systèmes, Vélizy-Villacoublay, France). After calibration, IMU primary accelerometer errors were ≤0.01 ± 0.04 g, secondary accelerometer errors were ≤0.05 ± 0.07 g, and gyroscope errors were ≤0.01 ± 0.01 rad/s (Appendix A).

### 2.2. Participants

Seventy-seven participants were recruited from the University of California, Davis, local running clubs, and the community at large. Participants were ≥18 years old and reported running ≥16.09 km per week for ≥6 months. Three participants were excluded from the analysis due to movement of an IMU (*n* = 2) or inability to complete the protocol as instructed (*n* = 1), leaving a final sample of 74 (32 males; 42 females; 0 non-binary; age 28 ± 12 years; Figure 1). All participants provided written informed consent, and procedures were approved by the University of California, Davis Institutional Review Board.

### 2.3. Protocol

Participant mass, height, and distance of the left and right lateral malleolus, fibular head, lateral epicondyle, and superior aspect of greater trochanter from the ground were measured. Using adhesive-bonded hook-and-loop fasteners, IMUs were attached to neoprene belts with anti-slip silicone inners, then wrapped with elastic straps as tightly as possible, within the limit of participant comfort. IMUs were placed anterior and superior to the lateral malleoli (*shank*), on the superior aspect of the iliac crests in line with the greater trochanter (*hip*), and on the superior aspect of the sacrum in line with the spine (*sacrum*) (Figure 2). 

Participants wore their own shoes and ran a 25 m runway with an embedded force plate (Kistler 9281, Kistler Group, Winterthur, Switzerland; 1000 Hz). The running speed was recorded using two custom-built laser speed gates, placed 2.5 m on each side of force plate center. Participants warmed up and practiced striking the force plate three times per side at their *slowest* (“the slowest pace you would use on a run”), *typical* (“the pace you use for the majority of your running”), and *fastest* (“the fastest pace you would use on a run”) self-selected speeds (Figure 3). During this warmup, markers on the lateral calcaneus and base of the fifth metatarsal were recorded using a conventional video camera (Exilim EX-FH25, Casio, Shibuya City, Tokyo, Japan; 120 Hz). Foot strike angle was calculated by subtracting a neutral standing foot angle from the foot angle at initial contact (Kinovea 0.9.5). Positive values indicate a more dorsiflexed foot at initial contact with values > 0.14 radians corresponding to rear-foot strike, −0.03 to 0.14 radians to mid-foot strike, and <−0.03 radians to forefoot strike [83]. After warm-up, five stances per side were collected at each speed for two surface conditions: (1) with a *track* surface covering the runway and force plate, and (2) with no covering on the hardwood *floor* of a basketball court. Participants always progressed from their slowest to fastest speeds, but the order of foot and surface was pseudo-randomized.

IMU data were synchronized within 100 ns of each other with a wireless network hub (Advanced Inertia Gateway, Inertia Technology, Enschede, The Netherlands). This hub sent voltage pulses that were synchronously recorded by IMU software (Inertia Studio v3.5.0, Inertia Technology, Enschede, The Netherlands) and a custom MATLAB script that simultaneously recorded speed and force data (R2018b, MathWorks, Natick, MA, USA). Pulse trains were cross correlated to synchronize signals. During data processing, we observed small timing discrepancies caused by the initialization of discrete MATLAB data acquisitions and small variances between the sampling rates of the IMU and MATLAB systems. Although extremely small, these discrepancies could accumulate over the course of the ~60 min data collection, leading to timing differences between the first and last synch events of a data collection (on the order of 10 s of ms). To ensure the input acceleration data were perfectly matched with the target force data, a conservative approach was used and only trials containing a synch event were analyzed (642 of 4440 trials). All other trials were discarded to ensure millisecond-level accuracy.

### 2.4. IMU Data Processing

Calibration matrices were applied to the IMU data. Quiet periods were identified (angular velocity < 0.5 rad/s and jerk < 0.01 m/s^3^ for at least 100 ms) and used to remove biases. Saturated frames from the primary accelerometer ($\left|a\right|$ > 15.5 g) were replaced with corresponding frames from the secondary accelerometer. Data were filtered with a 4th-order 50-Hz low-pass Butterworth filter. Angular velocity was drift-corrected using a Madgwick filter [84,85,86,87]. Starting at each quiet period, accelerations were used to estimate the IMU position in the inertial reference frame, then angular velocities were used to estimate frame-by-frame changes in IMU orientation and remove the gravity component from accelerations [88]. Data were then expressed in a segment coordinate system based on the principal component that explained the most variance in angular velocity during running (the medial-lateral axis) and the gravity vector during quiet standing [89,90]. IMU data during stance were extracted based on the instant the time-synchronized vertical force crossed a 10 N threshold. For more detailed IMU processing, see Appendix A.

### 2.5. Force Data Processing

Force data were filtered with a 4th-order 50-Hz low-pass Butterworth filter. A vertical force threshold of 10 N was used to define the start and end of stance. The first (or ‘impact’) peak was identified by performing a Fourier transform on the vertical GRF, then reconstructing a time domain signal from the ≥ 10 Hz high frequency (‘HiF’) components with an inverse Fourier transform [91]. First peak magnitude was defined as the magnitude of the original vertical GRF signal at the time when the HiF signal achieved its earliest peak occurring after 5% of stance duration. The loading rate was calculated from 20 to 80% of stance onset to first peak [92]. The second (or ‘active’) peak was defined as the maximum magnitude of the vertical GRF (or the magnitude of the second peak if two peaks were present). The average was also calculated across stance. These methods are depicted for a single stance in Figure 4.

### 2.6. Analysis

The first peak, loading rate, second peak, average force, and force time series were estimated with each capable method (Table 2). For methods that required a model to be built (see Appendix A), these features were estimated using a leave one out analysis where 74 models were iteratively trained with data from 73 participants, then used to estimate features for the single participant the model was not trained on. Errors were calculated by subtracting the ground truth force plate value from the estimated value (for first peak, loading rate, second peak, or average) or by calculating the RMSE (for time series).

One sample *t*-tests were used to compare each method’s error to the gold standard (0 error). Significance was set at *p* ≤ 0.05 with a false discovery rate (FDR) procedure to correct for multiple comparisons.

To account for the non-independence of the data (642 trials from 74 participants) and ensure proper estimation of variance, an adaptation of the Bland–Altman method was used [93]. Errors were entered into linear mixed-effects models in R (v4.2.2; R Foundation for Statistical Computing, Indianapolis, IN, USA) as described in Carstensen et al.’s approach to linked replicates [94,95]:(1)$$y_{method,participant,trial}=a_{method}+b_{participant}+C_{method,participant}+{D_{participant,trial}+\varepsilon }_{method,participant,trial}$$
where y corresponds to the model estimated error, lower case terms correspond to fixed effects, upper case terms correspond to random effects, and ε corresponds to error. Model assumptions of independence, normality, and homoscedasticity were validated by plotting within-participant variances against within-participant means, histograms of residuals, residuals for each level of random effect, and residuals as a function of fitted value. Method-specific variance components were extracted using the ‘MethComp’ package for R [96]:(2)$$C_{method,participant}\;\sim \;N(0,\tau_{method}^{2})$$
(3)$$\varepsilon_{method,participant,trial}\;\sim \;N(0,\sigma_{method}^{2})$$
where the values of C and ε are normally distributed about zero with variances of $\tau_{method}^{2}$ and $\sigma_{method}^{2}$ for each method. This allowed for the estimation of: (1) method biases that quantify accuracy (mean error); (2) repeatability coefficients (RC) that quantify the largest absolute difference predicted between two measurements on the same participant under identical circumstances; and (3) limits of agreement (LOA) that quantify precision (limits within which 95% of future errors for a given method are expected to fall), using the equations:(4)$${RC}_{method}=\pm \;2.83\sigma_{method}$$
(5)$${LOA}_{method}=\pm \;1.96\sqrt{\tau_{0}^{2}+\tau_{method}^{2}+\sigma_{0}^{2}+\sigma_{method}^{2}}$$
where $\tau_{0}^{2}$ and $\sigma_{0}^{2}$ correspond to variances for the gold standard.

To evaluate if any potential explanatory variables affected error, a second set of linear mixed effects models was developed for each method. These models added surface, speed, and foot strike angle as fixed effects. A *p* ≤ 0.05 for any fixed effect was interpreted as that fixed effect accounting for a significant amount of a method’s error (i.e., model-estimated force was significantly affected by the running surface, speed, and/or foot strike angle).

## 3. Results

### 3.1. First Peak

Using Carstensen’s method for linked replicates [94], biases, RCs, and LOAs were calculated for each method capable of first peak estimation (Figure 5; Table 3). First peak magnitude was estimated at the shank by one method, which was among the best performing (biases < 200 N): ‘Higgins shank’ (+16.46 ± 879.51 N or +1.35 ± 72.23%; bias ± LOA). At the hip by ‘Kiernan hip’ (−43.18 ± 957.18 N or −3.55 ± 78.61%) and ‘Higgins hip’ (+16.46 ± 906.02 N or +1.35 ± 74.41%). At the sacrum by nine methods, with four among the best performing: ‘Kim displacement’ (−132.74 ± 823.12 N or −10.90 ± 67.60%), ‘Kiernan sacrum’ (−33.81 ± 961.22 N or −2.78 ± 78.94%), ‘Pogson’ (−101.25 ± 839.57 N or −8.32 ± 68.95%), and ‘Pogson xynorm’ (−115.06 ± 850.78 N or −9.45 ± 69.87%).

A second set of linear mixed effects models were performed on each of the best performing methods (biases < 200 N) to examine the role of running speed, surface, and foot strike angle as potential explanatory variables. These models revealed that error in each of the best performing methods was significantly explained by running speed and foot strike angle (*p* values ≤ 0.05) but not by running surface (*p* values > 0.05) (Table 3). To illustrate these effects, model-predicted biases were plotted as a function of speed and foot strike angle (Figure 6). Yellow, red, and purple colors in these plots correspond to overestimations (positive errors where the estimate has a greater magnitude than the gold standard), while darker blue colors correspond to underestimations (negative errors where the estimate has a smaller magnitude than the gold standard). All methods showed the same general pattern with overestimates at low speeds and foot strike angles, and underestimates at high speeds and foot strike angles.

### 3.2. Loading Rate

Three methods were capable of estimating loading rate from shank accelerations and were among the best performing methods (biases < 10 kN/s): ‘Veras shank res’ (+4.37 ± 62.56 kN/s or +8.26 ± 118.16%; bias ± LOA), ‘Veras shank y’ (−3.25 ± 59.96 kN/s or −6.13 ± 113.25%), and ‘Higgins shank’ (−1.71 ± 56.69 kN/s or −3.23 ± 107.08%). Three were capable of estimating loading rate from hip acceleration but only ‘Higgins hip’ was among the best performing (+0.02 ± 52.36 kN/s or +0.04 ± 98.90%). Ten methods were capable of estimating loading rate from sacrum acceleration, of these, the best performing were: ‘Kim displacement’ (−6.34 ± 50.52 kN/s or −11.98 ± 95.43%), ‘Veras sacrum res’ (−1.95 ± 49.34 kN/s or −3.68 ± 93.19%), ‘Veras sacrum y’ (−1.04 ± 50.52 kN/s or −1.96 ± 95.42%), ‘Pogson’ (−5.32 ± 51.82 kN/s or −10.04 ± 97.87%), and ‘Pogson xynorm’ (−7.83 ± 51.21 kN/s or −14.80 ± 96.73%) (Figure 7).

The linear mixed effects models examining the role of running speed, surface, and foot strike angle on the best performing methods (biases < 10 kN/s) revealed that error in all methods was significantly explained by running speed and foot strike angle (*p* values ≤ 0.05) but not by running surface (*p* values > 0.05) (Table 4). All methods showed a similar pattern of overestimating loading rates at low speeds and foot strike angles and underestimating at high speeds and foot strike angles (Figure 8).

**Table 4 sensors-23-08719-t004:** Best performing loading rate estimation methods. Overall performance shown as biases (accuracy), RCs (repeatability), and LOAs (precision) color-coded from the absolute minimum (green) to absolute maximum (purple) values observed within-column. Biases that significantly differed from 0 (*p* ≤ 0.05 with FDR correction) are marked *. Performance across conditions shown as coefficients for the intercept of surface (added to model-estimated error for the track condition but not the floor condition) and slopes for running speed (in m/s) and foot strike angle (in radians). If surface, speed, or foot strike explains a significant (*p* ≤ 0.05) amount of error, it is highlighted pink and marked *.

	Overall Performance	Performance across Conditions
Method	Bias (kN/s)	RC (kN/s)	LOAs (kN/s)	Speed	Surface	Foot Strike
Veras shank res	+4.37 *	36.96	62.56	−8.28 *	−0.29	−36.00 *
Veras shank y	−3.25 *	32.96	59.96	−9.14 *	−0.52	−33.12 *
Higgins shank	−1.71	29.99	56.69	−3.47 *	0.92	−16.17 *
Higgins hip	+0.02	20.58	52.36	−10.66 *	−0.61	−21.41 *
Kim displacement	−6.34 *	11.98	50.52	−6.90 *	−1.45	−37.74 *
Veras sacrum res	−1.95	9.47	49.34	−12.36 *	−0.96	−22.08 *
Veras sacrum y	−1.04	10.18	50.52	−13.83 *	−0.96	−27.55 *
Pogson	−5.32 *	16.78	51.82	−5.98 *	−1.04	−40.57 *
Pogson xynorm	−7.83 *	17.25	51.21	−5.99 *	−1.03	−42.16 *
	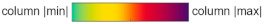			

**Figure 8 sensors-23-08719-f008:**
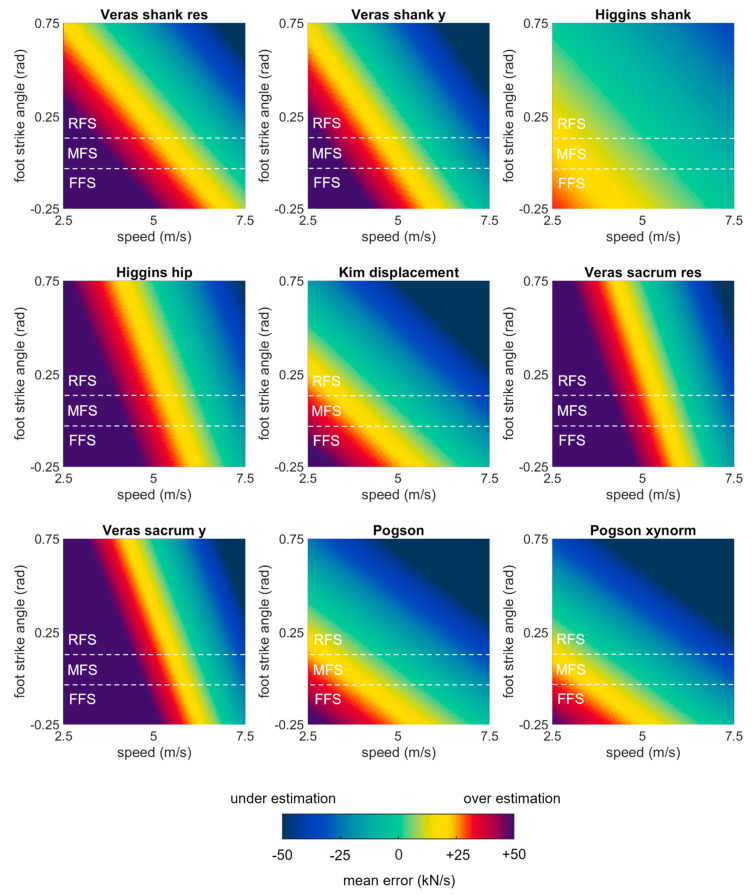
Bias in loading rate estimates predicted by mixed effects models for each of the best performing methods, plotted as a function of speed and foot strike angle. Green values represent perfect agreement with the gold standard; yellow, red, and purple values represent positive biases (overestimates); darker blue values represent negative biases (underestimates). Foot strike angles corresponding to rear-, mid-, and fore-foot strike patterns are labelled (RFS, MFS, and FFS) and divided with dashed white lines.

### 3.3. Second Peak

Four methods could estimate second peak from shank accelerations, all were among the best performing methods (biases < 100 N): ‘Charry’ (+58.48 ± 478.42 N or +3.53 ± 28.85%; bias ± LOA), ‘Thiel’ (−90.32 ± 1162.97 N or −5.45 ± 70.12%), ‘Veras shank res’ (−98.67 ± 470.96 N or −5.95 ± 28.40%), and ‘Veras shank y’ (−88.75 ± 471.45 N or −5.35 ± 28.43%). Five methods could estimate second peak from hip accelerations, with two among the best performing: ‘Neugebauer’ (+14.04 ± 488.09 N or +0.85 ± 29.43%) and ‘Kiernan hip’ (−7.56 ± 572.04 N or −0.46 ± 34.49%). At the sacrum, 16 methods could estimate second peak, with seven among the best performing: ‘Kim acceleration’ (−27.21 ± 721.47 N or −1.64 ± 43.50%), ‘Kim displacement’ (+20.68 ± 696.86 N or +1.25 ± 42.02%), ‘Kiernan sacrum’ (−4.20 ± 563.80 N or −0.25 ± 33.99%), ‘Veras sacrum y’ (−74.18 ± 492.08 N or −4.47 ± 29.67%), ‘Wundersitz 20 Hz’ (+34.19 ± 1089.85 N or +2.06 ± 65.71%), ‘Pogson’ (+25.36 ± 745.01 N or +1.53 ± 44.92%), and ‘Pogson xynorm’ (−2.39 ± 730.18 N or −0.14 ± 44.02%) (Figure 9).

The linear mixed effects models examining the role of running speed, surface, and foot strike angle on the best performing methods revealed that error in all methods was significantly explained by running speed (*p* values ≤ 0.05). Foot strike angle also significantly explained error in eight of 13 methods (*p* values ≤ 0.05) but not in ‘Kim acceleration’, ‘Kim displacement’, ‘Wundersitz 20 Hz’, ‘Pogson’, and ‘Pogson xynorm’ (*p* values > 0.05). Surface explained significant variation in only one method: ‘Wundersitz 20 Hz’ (all other *p* values > 0.05) (Table 5). Most methods showed the same general pattern with overestimates at low speeds and high foot strike angles. There were two exceptions: ‘Thiel’ had underestimates at low speeds and foot strike angles and overestimates at high speeds and foot strike angles, while ‘Pogson’ showed generally stable performance with a small (but significant) increase in error at faster speeds (Figure 10).

### 3.4. Average Force

Eight sacrum methods could calculate the average force from an estimated time series, the four best performing (biases < 100 N) were: ‘Kim acceleration’ (−67.76 ± 381.52 N or −6.87 ± 38.68%; bias ± LOA), ‘Kim displacement’ (+7.66 ± 394.37 N or +0.78 ± 39.98%), ‘Pogson’ (−3.18 ± 367.81 N or −0.32 ± 37.29%), and ‘Pogson xynorm’ (−4.87 ± 278.80 N or −0.49 ± 28.26%) (Figure 11).

The linear mixed effects models examining the role of running speed, surface, and foot strike angle on each of the best performing methods revealed that error in all methods except ‘Pogson’ was significantly explained by running speed (*p* values ≤ 0.05). Foot strike angle significantly explained error in only ‘Pogson’ (all other *p* values > 0.05). Surface significantly explained error in only ‘Pogson xynorm’ (all other *p* values > 0.05) (Table 6). Three methods showed the same general pattern with overestimates at low speeds. ‘Pogson’ was an exception, showing generally stable performance with a small (but significant) increase in error at lower foot strike angles (Figure 12).

### 3.5. Time Series

Eight sacrum methods could estimate force time series, of these the four best performing (RMSEs < 250 N) were: ‘Kim acceleration’ (245.69 ± 212.85 N or 24.91 ± 21.58%; mean ± LOA), ‘Kim displacement’ (240.43 ± 215.57 N or 24.37 ± 21.85%), ‘Pogson’ (237.33 ± 218.09 N or +24.06 ± 22.11%), and ‘Pogson xynorm’ (180.32 ± 230.62 N or 18.28 ± 23.38%) (Figure 13).

The linear mixed effects models examining the role of running speed, surface, and foot strike angle on each of the best performing methods (RMSEs < 250 N) revealed that error in all methods was significantly explained by running speed (*p* values ≤ 0.05) but not by foot strike angle or running surface (*p* values > 0.05) (Table 7). All methods showed the same general pattern with higher predicted RMSEs at higher speeds but ‘Pogson xynorm’ had lower predicted mean RMSEs than any other method across its entire range (Figure 14).

The ‘Kim acceleration’ and ‘Kim displacement’ methods produced highly stereotyped time series with little variation (i.e., regardless of the input acceleration, the output force estimate was similar) (Figure 15). This caused relatively large errors, particularly at the locations of the first and second peak. In contrast, ‘Pogson xynorm’ had much more variation in its estimated values and better fit the data (lower errors), however, errors remained high around the first peak.

## 4. Discussion

Performance was evaluated for 27 methods of estimating vertical GRF during running from a single wearable accelerometer on the shank or approximate COM. For each method, forces were estimated from 74 runners across two different surfaces (wood floor, running track), three self-selected speeds (slowest, typical, fastest), and a range of foot strike angles (including fore-, mid-, and rear-foot strike patterns). Errors were quantified as the difference between the estimated and ground truth forces. 

Based on the observed errors, we recommend the ‘Pogson’ or ‘Pogson xynorm’ methods for several reasons: First, these methods use a single accelerometer on the sacrum to estimate bilateral forces. This is advantageous over shank and hip methods that either (a) do not allow for the estimation of bilateral forces, (b) require an assumption of bilateral symmetry, or (c) require two accelerometers. Second, in contrast to most other methods, these methods can estimate every feature of the vertical GRF investigated here (first peak, loading rate, second peak, average, and time series). Third, these methods had relatively stable performance across speeds, foot strike angles, and running surfaces. Fourth, these methods were consistently high performers (had low biases, RCs, and LOAs) for the second peak, average, and time series estimation. Potential users should, however, balance their own design needs when choosing a method. For example, if estimating the second peak under known speed and foot strike conditions, other methods may have similar accuracy but better reliability and precision (e.g., the ‘Neugebauer’, ‘Kim’, or ‘Kiernan’ methods).

For first peak and loading rate estimation, the ‘Pogson’ methods were outperformed by other methods (e.g., ‘Higgins’, ‘Kiernan’, or ‘Veras’). However, this does not affect our overall recommendation due to the high LOAs observed for the first peak and loading rate estimation: Even the best performing methods had LOAs exceeding ±67.60 and ±93.19% of first peak and loading rate target magnitudes. These LOAs are likely larger than any potential between-group effects or within-participant changes, suggesting that first peak and loading rate cannot be estimated with sufficient precision with any of the methods investigated here. Given these results, we do not currently recommend the estimation of first peak or loading rate from accelerometers.

Despite the poor results for first peak and loading rate estimation, the observed biases for all estimated vertical GRF features were at or below those originally reported for nine of the 13 publications from which methods were derived [48,54,63,73,75,76,77,78,79]. Some of the error we did observe was attributable to speed, surface, and foot strike angle. Thus, one approach to decrease error may be to include these explanatory variables as model inputs. For example, Alcantara et al. [16] added speed, slope of the running surface, and foot strike pattern to their force prediction model and reported RMSEs of 106.78 N (or 6.4%) for time series estimation. These results are an improvement over the methods recommended here (although direct comparison is difficult as Alcantara et al. included the prediction of zero forces during swing in their RMSE calculations, which may have reduced their errors relative to our analysis of stance phase only when forces are non-zero) (see also: [56,68,79]). Thus, including speed, foot strike angle, and surface may improve the performance of future methods, provided that these variables can be quantified precisely and accurately in the field (e.g., [97,98,99]).

The inclusion of other explanatory variables may also improve performance. Thirteen of the 27 methods investigated here estimated force using linear regressions that included anthropometrics (mass, height, and/or leg length), sex, and/or age as explanatory variables [40,48,63,73,77,78,79]. Although these methods did not receive our final recommendation, many had very reasonable results. These results are particularly salient in contrast to the poor performance of methods that estimated force by simply multiplying acceleration by mass without additional explanatory variables [54,55,75,80]. This performance difference suggests that including explanatory variables improved the performance and may be warranted in future models.

Reductions in error could also be achieved by fitting participant-specific models. For example, Kiernan et al. [79] reported that including a random effect of participants reduced the error in their model by ~40% (see also: [40,63,78]). Thus, participant-specific models likely offer more accuracy, reliability, and precision than participant-general models and should be used when time and resources allow. Participant-specific models were not examined here to ensure that the methods studied were broadly applicable and accessible. This design choice maximizes their in-field utility and allows them to be applied to novel participants without the necessity of taking gold-standard measurements in-lab and using computational resources to develop a model for each participant.

Conversely, the choice to use acceleration data that were time synchronized to target force data with millisecond accuracy stands in contrast to the goal of applying these methods in-field. This choice was made to ensure that errors in accelerometer-based stance identification did not affect our results and lead to erroneous conclusions. When taking measures outside the lab, however, the use of time synchronized force plate data is not available. Currently, even the best methods to identify stance have errors of −30.4 ± 118.8 and −2.8 ± 149.9 ms (bias ± LOA) for initial and terminal contact, respectively [15]. These errors may interfere with accurate segmentation of acceleration data during stance and thus, until more exact acceleration-based stance identification is possible, acceleration signals will vary in duration relative to their target force signals. It is likely that the regression methods that used peak acceleration values to estimate discrete force variables are more robust to these discrepancies in duration [40,48,63,73,77,78,79]. The peaks used by these methods tend to be greater than the acceleration values immediately pre- or proceeding the initial and terminal contact and tend to be ~mid-stance, so are unlikely to be removed accidentally (see Appendix A). In contrast, methods attempting to estimate continuous force time series are likely vulnerable to errors in input data duration. 

To explore this, errors for the recommended ‘Pogson xynorm’ method were recalculated using acceleration input data that were stance-segmented based on acceleration signal features instead of a force plate threshold (using the ‘Auvinet’ stance identification method [100] as implemented by Kiernan et al. [15]) In contrast to the expectation that errors would increase, they were comparable to those observed when segmenting based on the time synchronized force signal (first peak: −104.75 ± 876.71 N; loading rate: −4.71 ± 53.31 kN/s; second peak: −5.03 ± 683.64 N; average: −5.83 ± 341.02 N; time series: 178.25 ± 207.58 N;). Model performance may have been maintained due to providing a more stereotyped input (i.e., acceleration data segmented based on acceleration features) and/or due to providing a larger set of training data (since millisecond accuracy was not required, all 4440 trials were used vs. the 642 exactly synchronized trials available for the other methods). In any case, the ‘Pogson_Auvinet’ method’s performance demonstrates the promise of current methods to estimate vertical GRF second peak, average, and time series in the field.

Before applying these force estimation methods in the field, potential users should consider the conditions and participants used to develop and validate these methods. It should not be assumed that these methods will work under other conditions or for other participants. For example, only over-ground running on two level surfaces was quantified here, so it should not be assumed that results will hold for incline/decline running, treadmill running, or running on other surfaces (e.g., sand, grass/turf, asphalt, concrete). That said, consistent with previous findings that changes in surface do not affect vertical GRF [101], results demonstrated that surface rarely explained error, suggesting that the estimated forces were robust to changes in surface. Potential users should also consider that the current sample represents a relatively homogenous group of runners (Figure 1). Thus, if studying participants drawn from different populations, these results may not be representative, and the included code/models may not produce estimations with the accuracy, reliability, and precision reported here.

Careful consideration should also be given to any differences in acceleration processing and/or coordinate conventions as any differences in acceleration inputs could affect force outputs. For example, the ‘Wundersitz’, ‘Meyer’, ‘Gurchiek’, and ‘Day’ methods [54,55,75,80] all used the same general approach of multiplying acceleration by mass to estimate force. Despite this common approach, results differed across these methods due to differences in acceleration processing. Expressing accelerations in different coordinate systems will also change the input and affect the output. All but two of the publications the methods were derived from used a WCS (Table 1). These coordinate systems assume alignment with segments or the inertial vector and may be particularly prone to altering acceleration inputs. For example, an accelerometer placed on the sacrum could deviate from its assumed inertial alignment due to lumbosacral curvature and adiposity or a participant leaning forward during running [102,103]. Any discrepancies in the placement of an accelerometer could also change the data. More consistent data may be obtained with SCS and TCCS. Thus, all analyses in this paper are presented in the SCS. Although not included here, analyses were repeated in the WCS and TCCS. In contrast to the *a priori* expectation that SCS and TCCS would outperform the WCS, a systematic effect of coordinate system was only found in methods that multiplied acceleration by mass with no other coefficients [54,55,75,80]. When unique model coefficients, weights, and/or biases were calculated for each coordinate system, there were no systematic differences between them (Appendix A). This result should, however, be interpreted with caution: each of the IMUs in this study was placed by the same experimenter, their positions were monitored throughout data collection, and any movement of the IMU led to elimination of a participant (*n* = 2). Thus, the WCS in this study is likely more consistent than under field conditions, where wearables may be placed across many repeated data collections by individuals with little training, leading to misalignment and inconsistency. Thus, we still caution against the use of a WCS.

## 5. Conclusions and Practical Applications

We recommend the ‘Pogson’ methods to estimate vertical ground reaction force second peak, average, and time series. We do not currently recommend the estimation of first peak or loading rate due to the large observed limits of agreement for these variables. For each method, code to automatically process stance-segmented accelerometer data is available at https://github.com/DovinKiernan/MTFBWY_running_vGRF_from_a, accessed on 12 September 2023. This code should be applied with careful consideration of the sample it was developed and validated on and the data processing applied to the acceleration inputs used to train the models. Future research should investigate whether the inclusion of anthropometrics, sex, age, field measures of speed and foot strike angle, or other explanatory variables can improve model performance. The results reported here should be used as a benchmark for the performance of future models and details on accuracy, reliability, and precision should be reported.

## Figures and Tables

**Figure 1 sensors-23-08719-f001:**
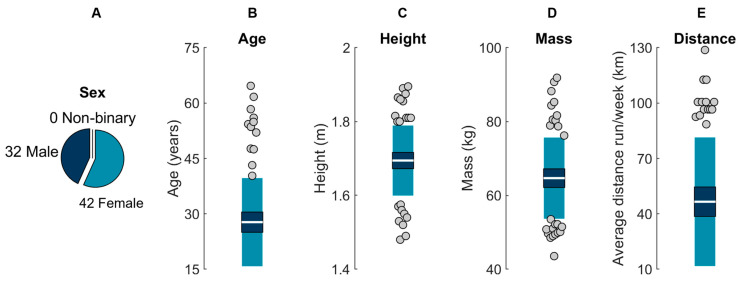
Participant (**A**) sex, (**B**) age, (**C**) height, (**D**) mass, and (**E**) self-reported average distance run per week. The white horizontal line represents the mean; dark blue represents ±95% confidence interval (±1.96 SEM) around the mean; and light blue represents ±1 SD around the mean. Gray dots represent participants outside ±1 SD.

**Figure 2 sensors-23-08719-f002:**
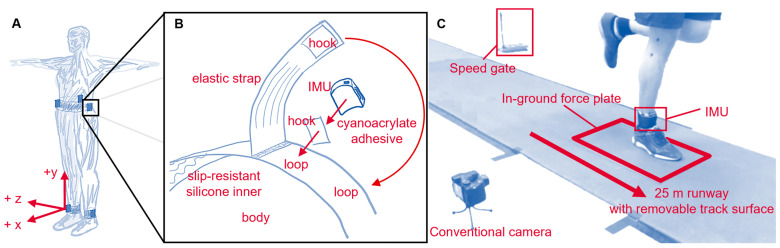
(**A**) IMU placement and coordinate conventions. For consistency, different conventions used across methods have been standardized to ISB conventions [82]: Segment coordinate systems (SCS) were defined as anterior (+x), proximal (+y), and medial-lateral (with right defined as +z); wearable coordinate systems (WCS) were defined square to the IMU housing, which was roughly aligned with the direction of progression (+x), longitudinal axis (+y proximal), and right (+z); tilt-corrected coordinate systems (TCCS) were defined as vertical (+y) and the projections of direction of progression (+x) and the medial-lateral axis (+z right) onto the horizontal plane. (**B**) Belt design and IMU fixation. (**C**) Experimental setup.

**Figure 3 sensors-23-08719-f003:**
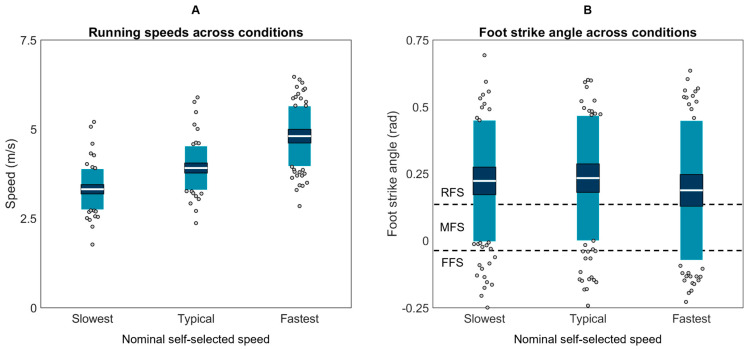
(**A**) Mean speeds and (**B**) foot strike angles used by each participant across the slowest, typical, and fastest running speed conditions. The white horizontal line represents the mean; dark blue represents ±95% confidence interval (±1.96 SEM) around the mean; and light blue represents ±1 SD around the mean. Gray dots represent participants outside ±1 SD. RFS is rear foot strike, MFS is mid foot strike, and FFS is fore foot strike.

**Figure 4 sensors-23-08719-f004:**
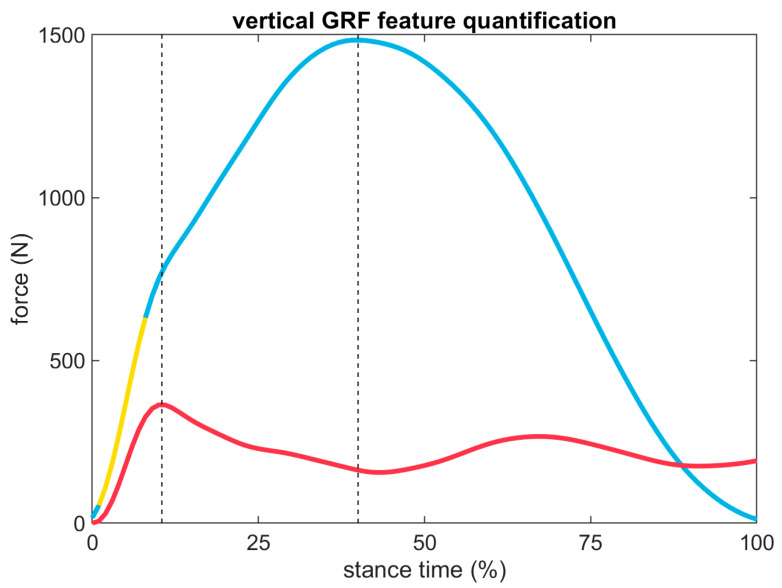
Vertical GRF (blue) and HiF reconstruction (red) for an example stance. Vertical dashed lines indicate the timing of first and second vertical GRF peaks. Note, although the first peak is difficult to visually identify in the original signal (blue), a consistent point in the HiF signal can still be identified (red). The yellow highlighted region from 20 to 80% of the first peak was used to calculate the loading rate.

**Figure 5 sensors-23-08719-f005:**
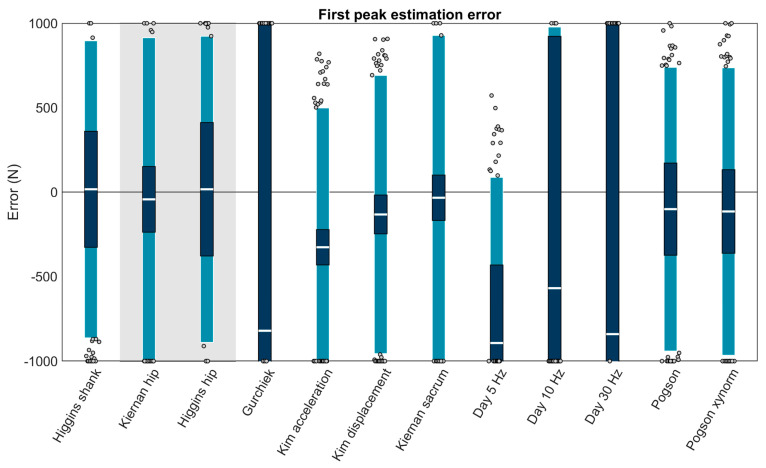
Bias (white bar), ±RC (dark blue), and ±LOA (light blue) in first peak estimation for each capable method. Gray dots represent trials falling outside the LOA. Values outside ± 1000 N are plotted at the axis limits. A value of 0 represents perfect agreement with the force plate. Positive values indicate the method overestimated the first peak. Negative values indicate the method underestimated the first peak. The method with a white background on the left is for accelerometers on the shank, methods with a gray background are for the hip, and methods with a white background on the right are for the sacrum.

**Figure 6 sensors-23-08719-f006:**
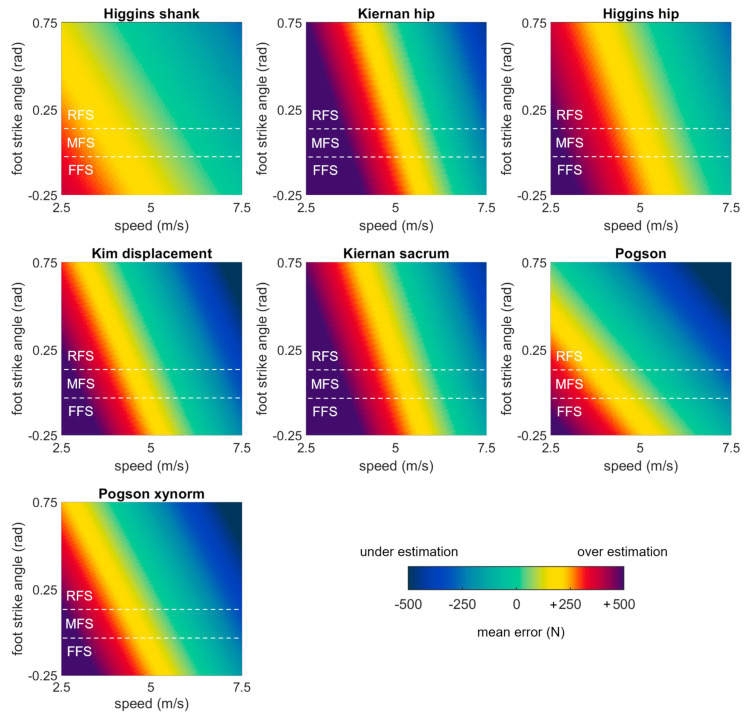
Bias in first peak estimates predicted by mixed effects models for each of the best performing methods, plotted as a function of speed and foot strike angle. Green values represent perfect agreement with the gold standard; yellow, red, and purple values represent positive biases (overestimates); darker blue values represent negative biases (underestimates). Foot strike angles corresponding to rear-, mid-, and fore-foot strike patterns are labelled (RFS, MFS, and FFS) and divided with dashed white lines.

**Figure 7 sensors-23-08719-f007:**
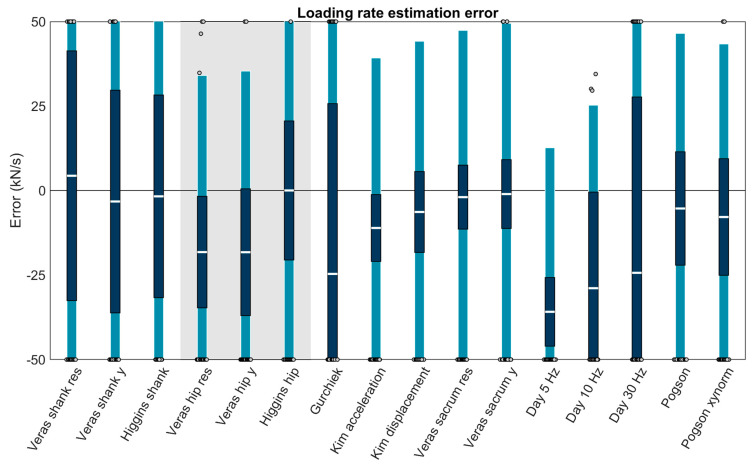
Bias (white bar), ± RC (dark blue), and ± LOA (light blue) in the loading rate for each capable method. Gray dots represent trials falling outside the LOA. Values outside ± 50 kN/s are plotted at the axis limits. A value of 0 represents perfect agreement with the force plate. Positive values indicate the method overestimated loading rate. Negative values indicate the method underestimated loading rate. Methods with a white background on the left are for accelerometers on the shank, methods with a gray background are for the hip, and methods with a white background on the right are for the sacrum.

**Figure 9 sensors-23-08719-f009:**
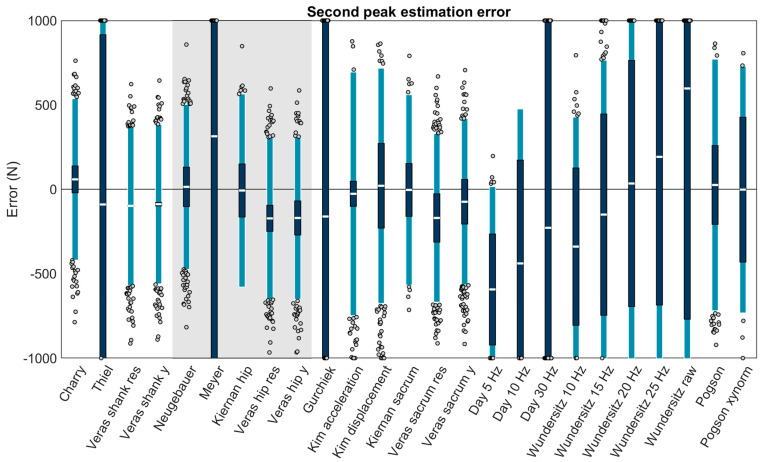
Bias (white bar), ± RC (dark blue), and ± LOA (light blue) in the second peak estimation for each capable method. Gray dots represent trials falling outside the LOA. Values outside ± 1000 N are plotted at the axis limits. A value of 0 represents perfect agreement with the force plate. Positive values indicate the method overestimated second peak. Negative values indicate the method underestimated second peak. Methods with a white background on the left are for accelerometers on the shank, methods with a gray background are for the hip, and methods with a white background on the right are for the sacrum.

**Figure 10 sensors-23-08719-f010:**
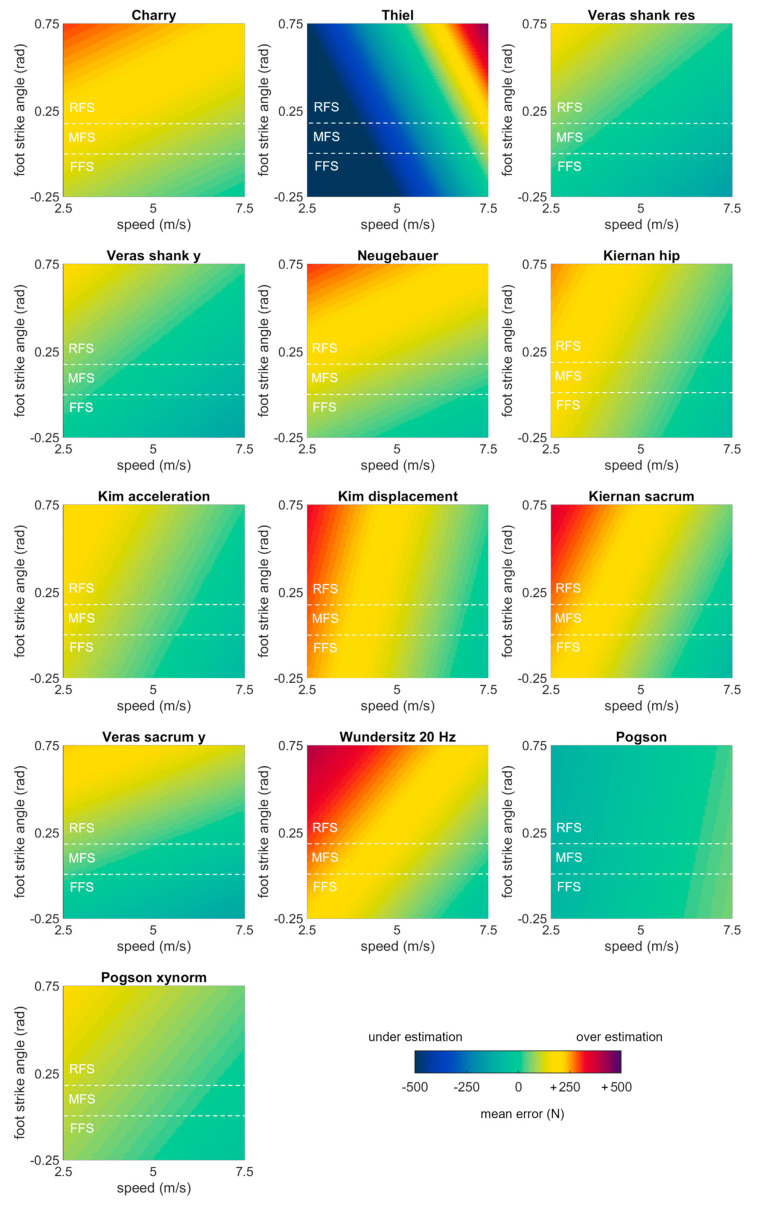
Bias in second peak estimates predicted by mixed effects models for each of the best performing methods, plotted as a function of speed and foot strike angle. Green values represent perfect agreement with the gold standard; yellow, red, and purple values represent positive biases (overestimates); darker blue values represent negative biases (underestimates). Foot strike angles corresponding to rear-, mid-, and fore-foot strike patterns are labelled (RFS, MFS, and FFS) and divided with dashed white lines.

**Figure 11 sensors-23-08719-f011:**
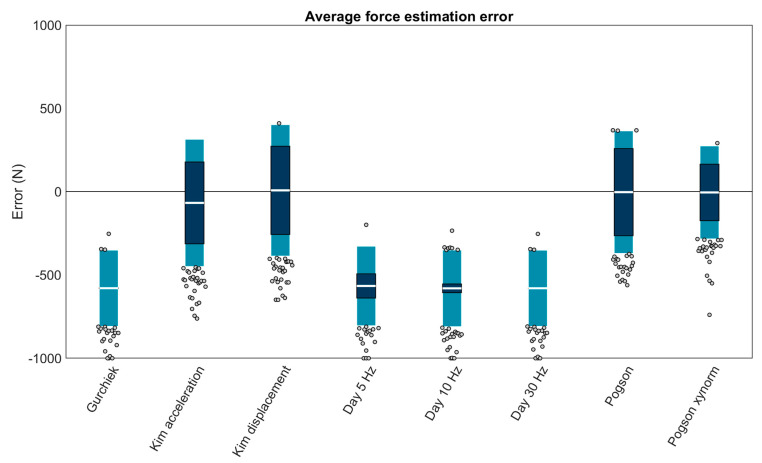
Bias (white bar), ± RC (dark blue), and ± LOA (light blue) in average force estimation for each capable method. Gray dots represent trials falling outside the LOA. Values outside ± 1000 N are plotted at the axis limits. A value of 0 represents perfect agreement with the force plate. Positive values indicate the method overestimated the average force. Negative values indicate the method underestimated the average force. All methods were for the sacrum.

**Figure 12 sensors-23-08719-f012:**
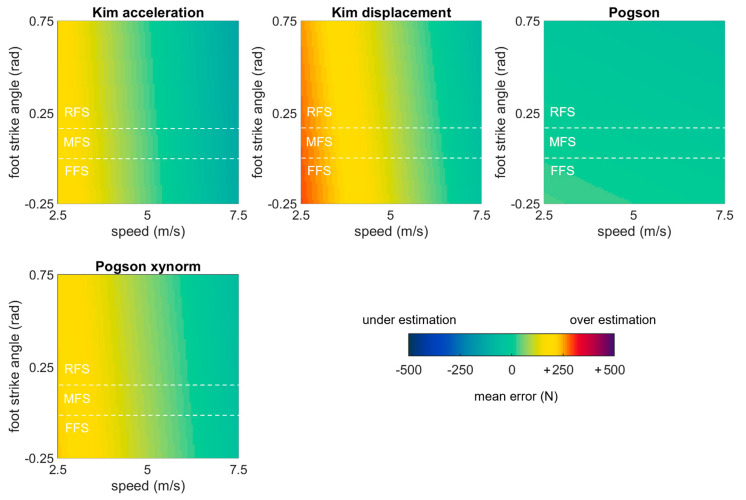
Bias in average force estimates predicted by mixed effects models for each of the best performing methods, plotted as a function of speed and foot strike angle. Green values represent perfect agreement with the gold standard; yellow, red, and purple values represent positive biases (overestimates); darker blue values represent negative biases (underestimates). Foot strike angles corresponding to rear-, mid-, and fore-foot strike patterns are labelled (RFS, MFS, and FFS) and divided with dashed white lines.

**Figure 13 sensors-23-08719-f013:**
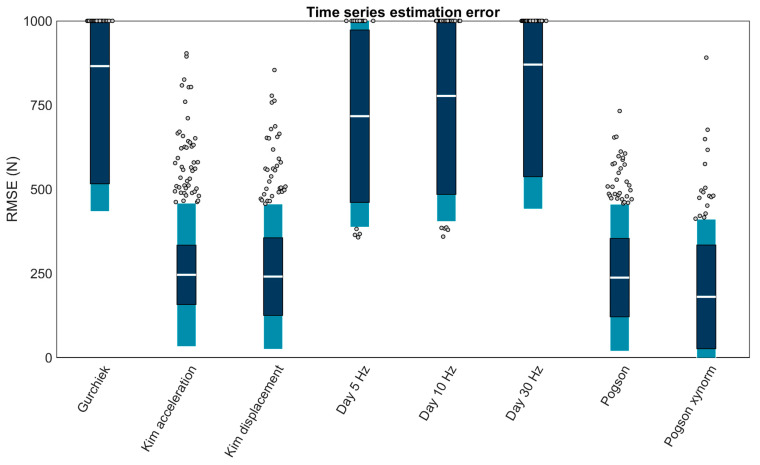
Mean RMSE (white bar), ± RC (dark blue), and ± LOA (light blue) in time series estimation for each capable method. Gray dots represent trials falling outside the LOA. Values > 1000 N are plotted at the axis limit. A value of 0 represents perfect agreement with the force plate. Positive values indicate larger errors. All methods were for the sacrum.

**Figure 14 sensors-23-08719-f014:**
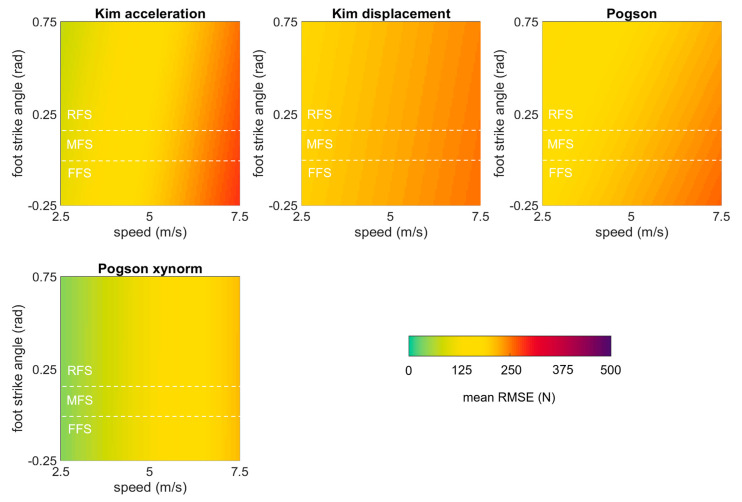
Mean RMSE in the estimated force time series predicted by mixed effects models for each of the best performing methods, plotted as a function of speed and foot strike angle. Green values represent perfect agreement with the gold standard; yellow, red, and purple values represent higher predicted mean RMSEs. Foot strike angles corresponding to rear-, mid-, and fore-foot strike patterns are labelled (RFS, MFS, and FFS) and divided with dashed white lines.

**Figure 15 sensors-23-08719-f015:**
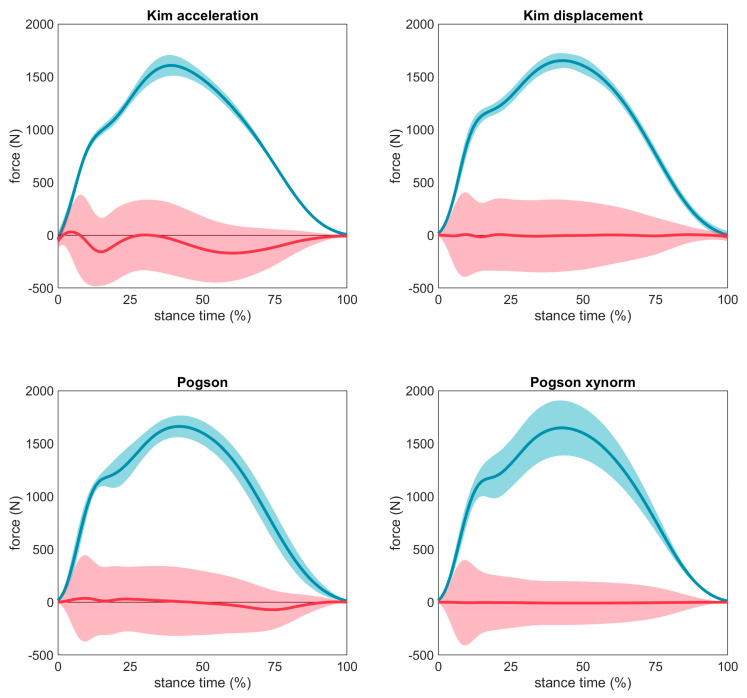
Mean time series estimated by each method (dark blue line) ± 1 SD (light blue shading) and the error between the gold standard time series (dark red line) ± 1 SD (light red shading). A red line at 0 indicates perfect agreement with the force plate, values above 0 indicate an overestimation of force at that time point, values below 0 indicate an underestimation of force at that time point.

**Table 1 sensors-23-08719-t001:** Thirteen publications met our inclusion criteria and presented one or more methods to estimate at least one feature of the vertical GRF (first peak, loading rate, second peak, average, or time series) from a single accelerometer on the shank or approximate COM (hip, sacrum, lower back, or upper back).

Publication	Sample	Foot-Strike	Speed	Surface	Placement	Signals	Range and Frequency	Targets	Ground Truth	Sync
Neugebauer 2012, 2014 [73,74]	*n* = 35 (20 F 15 M) children [73] *n* = 39 (20 F 19 M) injury free adults [74]	NR	2.2-3.9 m/s [73] 2.2–4.1 m/s [74]	90 [73] and 15 m [74] overground	Right iliac crest	$\alpha_{WCSres}$ [73] $\alpha_{WCSx,y}$ [74]	NR 40 Hz [73] ±6 g 100 Hz [74]	$F_{y,max}$	Force plate 1000 Hz	Average 30 [73] or 10 s [74]
Charry 2013 [75]	*n* = 3	NR	1.7–7.2 m/s	Overground	Medial mid-tibia	$\alpha_{WCSy}$	±24 g 100 Hz	$F_{y,second}$	Force plate 300 Hz	Video
Wundersitz 2013 [54]	*n* = 17 (5 F 12 M) uninjured team sport	NR	2.5–7.4 m/s	10 m overground	2nd Thoracic vertebra	$\alpha_{WCSy,res}$	±8 g 100 Hz	$F_{y,max}$	Force plate 100 Hz	Video
Meyer 2015 [76]	*n* = 13 (3 F 10 M) moderately active children	NR	1.7–2.8 m/s	10 m overground	Hip	$\alpha_{WCSy}$	±8 g 100 Hz	$F_{y,max}$	Force plate 2400 Hz	Average 8–15 steps
Gurchiek 2017 [55]	*n* = 15 (3 F 12 M)	NR	Sprinting and cutting	Overground	Sacrum	$\alpha_{GCSx,y,z}$	±24 g 450 Hz	$F_{y,t}$ $F_{y,average}$	Force plate 1000 Hz	Counter-movement jumps
Thiel 2018 [48]	*n* = 3 elite sprinters	NR	Sprint	Overground	Above medial malleolus	$\alpha_{WCSx,y,z}$	±16 g 250 Hz	$F_{y,max}$	Force plates 1000 Hz	LED flash
Kiernan 2020 [77]	*n* = 40 (NR)	NR	NR	25 m Overground	Sacrum; iliac crest	$\alpha_{SCSy}$	±100 g 1000 Hz	$F_{y,first}$ $F_{y,second}$	Force plate 1000 Hz	TTL pulses
Kim 2020 [78]	*n* = 7 (0 F 7 M)	NR	2.9 m/s	Treadmill	Sacrum	$\alpha_{WCSx,y,z}$	200 Hz	$F_{y,t}$	Force plate 400 Hz	NR
Pogson 2020 [79]	*n* = 15 (5 F 10 M) team sport players	NR	2.0–8.0 m/s	Overground	Back of upper torso	$\alpha_{WCSx,y,z}$	±16 g 100 Hz	$F_{y,t}$	Force plate 3000 Hz	Synchronous recording
Day 2021 [80]	*n* = 30 (21 F 9 M) NCAA Div 1 cross country	NR	3.8–5.4 m/s	Treadmill	Posterior waistband	$\alpha_{WCSx,y,z}$	500 Hz	$F_{y,t}$ $F_{y,max}$	Instrumented treadmill 500 Hz	Average 10 s (jump)
Higgins 2021 [40]	*n* = 30 (15 F 15 M) healthy	NR	~1.8–5.0 m/s	23 m overground	Superior to lateral malleolus; hip	$\alpha_{WCSy,res}$	±8 g 100 Hz	$F_{y,first}$ $\frac{dy}{dx}F_{y,first}$	Force plate 1000 Hz	Vertical jumps
Veras 2022 [63]	*n* = 131 (52 F 79 M) adults	NR	1.9–3.9 m/s	Treadmill	Superior to lateral malleolus; iliac crest; sacrum	$\alpha_{WCSy\&res}$	±16 g 100 Hz	$\frac{dy}{dx}F_{y,first}$ $F_{y,max}$	Instrumented treadmill 1000 Hz	Manual correction and cross-correlation

M—male; F—female; NR—not reported; α—acceleration; SCS—segment coordinate system; WCS—wearable coordinate system; GCS—global coordinate system (coordinate conventions defined below); $F_{y,first}$—first (or ‘impact’) peak; $\frac{dy}{dx}F_{y,first}$—loading rate to first peak; $F_{y,second}$—second (or ‘active’) peak; $F_{y,max}$—maximum, assumed to correspond to the second (or ‘active’) peak; $F_{y,t}$—time series.

**Table 2 sensors-23-08719-t002:** Twenty-seven methods were derived or adapted from the 13 publications in Table 1. Methods are sorted by accelerometer placement location. Methods that originally placed accelerometers on the lumbar or thoracic spine have been adapted to the sacrum. Not all methods could estimate all potential vertical GRF features. If a method was originally designed to estimate a feature, it is specified as ‘designed’ and highlighted in blue. If a method was designed to estimate a time series, all discrete vertical GRF features were then derived from that time series. These are marked ‘derived’ and highlighted pink.

		Estimated Force Variable				
Sensor Location	Method	First Peak	Loading Rate	Second Peak	Average	Time Series
Shank	Charry			designed		
	Thiel			designed		
	Veras shank res		designed	designed		
	Veras shank y		designed	designed		
	Higgins shank	designed	designed			
Hip	Neugebauer			designed		
	Meyer			designed		
	Kiernan hip	designed		designed		
	Veras hip res		designed	designed		
	Veras hip y		designed	designed		
	Higgins hip	designed	designed			
Sacrum	Gurchiek	derived	derived	derived	designed	designed
	Kim acceleration	derived	derived	derived	derived	designed
	Kim displacement	derived	derived	derived	derived	designed
	Kiernan sacrum	designed		designed		
	Veras sacrum res		designed	designed		
	Veras sacrum y		designed	designed		
	Day 5 Hz	derived	derived	designed	derived	designed
	Day 10 Hz	derived	derived	designed	derived	designed
	Day 30 Hz	derived	derived	designed	derived	designed
	Wundersitz 10 Hz			designed		
	Wundersitz 15 Hz			designed		
	Wundersitz 20 Hz			designed		
	Wundersitz 25 Hz			designed		
	Wundersitz raw			designed		
	Pogson	derived	derived	derived	derived	designed
	Pogson xynormed	derived	derived	derived	derived	designed

**Table 3 sensors-23-08719-t003:** Best performing first peak estimation methods. Overall performance shown as biases (accuracy), RCs (repeatability), and LOAs (precision) color-coded from the absolute minimum (green) to absolute maximum (purple) values observed within-column. Biases that significantly differed from 0 (*p* ≤ 0.05 with FDR correction) marked *. Performance across conditions shown as coefficients for the intercept of surface (added to model-estimated error for the track condition but not the floor condition) and slopes for running speed (in m/s) and foot strike angle (in radians). If surface, speed, or foot strike explains a significant (*p* ≤ 0.05) amount of error it is highlighted pink and marked *.

	Overall Performance	Performance across Conditions
Method	Bias (N)	RC (N)	LOAs (N)	Speed	Surface	Foot Strike
Higgins shank	+16.46	343.88	879.51	−83.70 *	11.89	−243.59 *
Kiernan hip	−43.18 *	194.73	957.18	−191.48 *	−5.69	−300.01 *
Higgins hip	+16.46	395.53	906.02	−131.03 *	−1.19	−231.61 *
Kim displacement	−132.74 *	115.26	823.12	−181.87 *	−12.86	−387.83 *
Kiernan sacrum	−33.81 *	134.60	961.22	−183.19 *	−11.26	−312.66 *
Pogson	−101.25 *	273.04	839.57	−135.50 *	−14.55	−535.86 *
Pogson xynormed	−115.06 *	247.59	850.78	−170.00 *	2.42	−427.28 *
	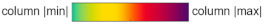			

**Table 5 sensors-23-08719-t005:** Best performing second peak estimation methods. Overall performance shown as biases (accuracy), RCs (repeatability), and LOAs (precision) color-coded from the absolute minimum (green) to absolute maximum (purple) values observed within-column. Biases that significantly differed from 0 (*p* ≤ 0.05 with FDR correction) are marked *. Performance across conditions shown as coefficients for the intercept of surface (added to model-estimated error for the track condition but not the floor condition) and slopes for running speed (in m/s) and foot strike angle (in radians). If surface, speed, or foot strike explains a significant (*p* ≤ 0.05) amount of error, it is highlighted pink and marked *.

	Overall Performance	Performance across Conditions
Method	Bias (N)	RC (N)	LOAs (N)	Speed	Surface	Foot Strike
Charry	58.48 *	79.70	478.42	−20.20 *	4.46	200.08 *
Thiel	−90.32 *	1006.17	1162.97	192.97 *	17.93	484.50 *
Veras shank res	−98.67 *	0.53	470.96	−30.82 *	3.04	182.12 *
Veras shank y	−88.75 *	10.07	471.45	−29.67 *	3.88	178.48 *
Neugebauer	14.04	116.89	488.09	−23.53 *	1.69	244.81 *
Kiernan hip	−7.56	157.16	572.04	−44.59 *	9.34	113.21 *
Kim acceleration	−27.21 *	74.26	721.47	−39.70 *	0.25	102.31
Kim displacement	20.68	250.96	696.86	−61.10 *	1.81	75.90
Kiernan sacrum	−4.20	156.73	563.80	−60.27 *	8.40	138.82 *
Veras sacrum y	−74.18 *	132.12	492.08	−22.25 *	4.19	268.09 *
Wundersitz 20 Hz	34.19	729.30	1089.85	−51.84 *	63.98 *	205.90
Pogson	25.36	233.50	745.01	28.92 *	2.74	−27.38
Pogson xynorm	−2.39	429.45	730.18	−23.62 *	18.74	93.89
	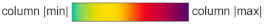			

**Table 6 sensors-23-08719-t006:** Best performing average force estimation methods. Overall performance shown as biases (accuracy), RCs (repeatability), and LOAs (precision) color-coded from the absolute minimum (green) to absolute maximum (purple) values observed within-column. Biases that significantly differed from 0 (*p* ≤ 0.05 with FDR correction) are marked *. Performance across conditions shown as coefficients for the intercept of surface (added to model-estimated error for the track condition but not the floor condition) and slopes for running speed (in m/s) and foot strike angle (in radians). If surface, speed, or foot strike explains a significant (*p* ≤ 0.05) amount of error, it is highlighted pink and marked *.

	Overall Performance	Performance across Conditions
Method	Bias (N)	RC (N)	LOAs (N)	Speed	Surface	Foot Strike
Kim acceleration	−67.76 *	245.71	381.52	−66.45 *	3.96	−24.06
Kim displacement	7.66	265.11	394.37	−67.73 *	5.20	−36.48
Pogson	−3.18	261.96	367.81	−5.48	3.68	−59.42 *
Pogson xynorm	−4.87	169.92	278.80	−54.79 *	14.12 *	−24.00
	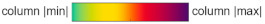			

**Table 7 sensors-23-08719-t007:** Best performing force time series estimation methods. Overall performance shown as biases (accuracy), RCs (repeatability), and LOAs (precision) color-coded from the absolute minimum (green) to absolute maximum (purple) values observed within-column. RMSEs that significantly differed from 0 (*p* ≤ 0.05 with FDR correction) are marked *. Performance across conditions shown as coefficients for the intercept of surface (added to model-estimated error for the track condition but not the floor condition) and slopes for running speed (in m/s) and foot strike angle (in radians). If surface, speed, or foot strike explains a significant (*p* ≤ 0.05) amount of error, it is highlighted pink and marked *.

	Overall Performance	Performance across Conditions
Method	RMSE (N)	RC (N)	LOAs (N)	Speed	Surface	Foot Strike
Kim acceleration	245.69 *	88.18	212.85	36.30 *	−3.45	−32.21
Kim displacement	240.43 *	115.29	215.57	11.64 *	4.84	−10.52
Pogson	237.33 *	116.38	218.09	16.84 *	−4.09	−40.05
Pogson xynorm	180.32 *	153.83	230.62	34.94 *	0.31	−0.85
	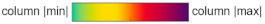			

## Data Availability

Data are not publicly available due to stipulations in the IRB protocol, however, all software has been made publicly available at
https://github.com/DovinKiernan/MTFBWY_running_vGRF_from_a accessed on 12 September 2023.

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
