# Peer review of "Acceleration-Based Estimation of Vertical Ground Reaction Forces during Running: A Comparison of Methods across Running Speeds, Surfaces, and Foot Strike Patterns"

_sensors, 2023, doi:10.3390/s23218719_

Round 1
Reviewer 1 Report
This manuscript compares 27 methods of estimating vertical ground reaction force first peak, loading rate, second peak, average, and/or time series from a single wearable accelerometer worn on the shank or approximate center of mass during running. On the whole, the research content of this manuscript is relatively meaningful, but there are also some problems that need to be modified.
1. There are too many keywords, it is recommended to keep them to 5.
2. At the end of section 1, Introduction, the overall article structure of the manuscript should be presented to facilitate readers' understanding of the manuscript.
3. In lines 120-126, it is recommended to include a detailed explanation of Table 2 in the main text paragraph. Other figures and tables in the text also have similar issues.
4. The parameters in some mathematical expressions in this manuscript should be explained.
5. The reviewer suggests that the sentences in the manuscript should use the simple present passive voice structure as much as possible, and it is also recommended to use "we" less frequently.
The language of this manuscript is generally acceptable, and it is recommended to use simple sentences as much as possible.
Author Response
- 
- There are too many keywords, it is recommended to keep them to 5.
 We have reduced the number of key words to four. - At the end of section 1, Introduction, the overall article structure of the manuscript should be presented to facilitate readers' understanding of the manuscript.
 Thank you for the suggestion. To facilitate readers’ understanding of the manuscript we have edited the section at the end of the introduction to streamline it and better preface the structure of the article. - In lines 120-126, it is recommended to include a detailed explanation of Table 2 in the main text paragraph. Other figures and tables in the text also have similar issues.
 We have added a more detailed explanation of Table 2. We have made similar additions for other tables that may have previously lacked detailed explanations. - The parameters in some mathematical expressions in this manuscript should be explained.
 Thank you for pointing out this oversight. We have fully explained all parameters in all equations. - The reviewer suggests that the sentences in the manuscript should use the simple present passive voice structure as much as possible, and it is also recommended to use "we" less frequently.
 The use of “we” has been reduced where possible but is retained in some areas where it conveys important subjectivity, e.g., “we recommend…” 

Reviewer 2 Report
The article presents the results of the verification of models of ground reaction forces during running determined on the basis of kinematic data. The article is very well prepared, with well-conducted literature research, which is the basis for research verifying various models of ground reaction forces determined on the basis of kinematic data. The methodology of the research, in which various models of ground reaction forces were verified, was also presented in great detail. The discussion of the research results is well conducted and is the basis for the final conclusions.
I recommend the article for publication. At the same time, I make a few small suggestions and comments:
1. The title of the work can be improved. The presented model of determining the reaction of the ground during the run can use kinematic data obtained from any measurement system, not only from accelerometric measurements.
2. It seems to me that the literature research conducted on the subject of the work could constitute a separate article.
Author Response
- The title of the work can be improved. The presented model of determining the reaction of the ground during the run can use kinematic data obtained from any measurement system, not only from accelerometric measurements.
Thank you for this recommendation, we have made the title more generalizable by changing “accelerometer-based” to “acceleration-based.”
- It seems to me that the literature research conducted on the subject of the work could constitute a separate article.
Although detailing the methods in the paper and supplement takes considerable space, we believe it is more convenient for readers and potential users to have access to all information in a single article.

Reviewer 3 Report
The main purpose of this paper is to estimate the force of a person during running by means of vertical ground reaction force. 27 different methods of estimating the vertical ground reaction force were compared through field exercises and testing of athletes, and finally the "Pogson" method was found to be the most accurate, reliable and precise.
1. The three diagrams of the devices on page 6 are not very clear; it would be better to introduce one diagram of a device in one paragraph, so that the reader can understand the route of operation of the device.
2. The statistical graph given in Figure 4 does not provide a detailed explanation as to why it is there, and the statistical graph is not analyzed and explained.
3. In the discussion of the results in section III, a large number of statistical graphs and tables are used to interpret the first and second peaks. The graphs feel heavily repetitive and can they be simplified.
4. The discussion in chapter IV is too lengthy, while the conclusions in chapter V are too short and should be properly balanced.
Author Response
- The three diagrams of the devices on page 6 are not very clear; it would be better to introduce one diagram of a device in one paragraph, so that the reader can understand the route of operation of the device.
The intent of Figures 2A-C is not to convey how the IMUs used in this study operate. Rather, these figures are intended to depict IMU placement and fixation and coordinate conventions (Fig 2 A and B) as well as to provide a snapshot of the data collection to facilitate readers’ understanding of the protocol (Fig 2 C). For details on the IMU specifications and operation beyond what are included in the paper we have added a link to the manufacturer’s documentation.
- The statistical graph given in Figure 4 does not provide a detailed explanation as to why it is there, and the statistical graph is not analyzed and explained.
We apologize for the lack of clarity, Figure 4 is not a statistical graph; rather, it depicts the method used to identify the different features of vertical ground reaction force investigated in this study (first peak, loading rate, and second peak). We have changed the text and figure caption to better convey this.
- In the discussion of the results in section III, a large number of statistical graphs and tables are used to interpret the first and second peaks. The graphs feel heavily repetitive and can they be simplified.
We appreciate the reviewers’ concern and have tried to balance presenting a concise and digestible results section with providing sufficient detail to understand the results. For the figures currently included in the paper, while the style of these is repetitive, the content is not. Estimations of five quantities were evaluated (first peak magnitude, loading rate, second peak magnitude, average, and time series). Each of these quantities is distinct and requires its own results (e.g., second peak estimation results do not tell the reader about first peak estimation results). For each quantity, a figure is required to compare the methods’ accuracy, repeatability, and precision, a second figure is required to describe the effects of foot strike angle and running speed on method accuracy, and a table is required to quantify the visuals presented in the figures. Thus, a total of 15 figures and tables is required (5 features x 3 per feature).
The repetition in figure style is intentional as it facilitates understanding: if a reader understands one figure, they can apply that understanding to all proceeding figures.
- The discussion in chapter IV is too lengthy, while the conclusions in chapter V are too short and should be properly balanced.
We believe this is a stylistic choice that fits the current paper. The Conclusions are intentionally short to facilitate a concise summary of the results and recommendations. The Discussion is longer to facilitate a more nuanced discussion of the results, limitations, and future directions in light of relevant research and to balance another reviewer’s concerns regarding its depth.

Reviewer 4 Report
Dear Editor,
Thank you very much for the opportunity to review the manuscript “Accelerometer-Based Estimation of Vertical Ground Reaction Forces During Running: A Comparison of Methods Across Running Speeds, Surfaces, and Foot Strike Patterns”. The objective of the manuscript was to compare 27 methods of estimating vertical ground reaction force first peak, loading rate, second peak, average, and/or time series from a single wearable accelerometer worn on the shank or approximate center of mass during running. The sample consisted of 74 people across different running surfaces, speeds, and foot strike angles and biases. After reading the manuscript, I realized that:
1 – The introduction is well structured. However, I make the following suggestions: 1 – Remove the statement “In this paper we compare 27 methods of estimating vertical ground reaction forces (GRFs) from a single wearable accelerometer during running. GRFs are external reaction forces created with equal magnitude and opposite sense to the force that the foot applies to the ground with each step.”. It's meaningless. This is the purpose of the manuscript and should not appear in the first paragraph. Authors should review how to begin the introduction. 2 – Explain the objective at the end of the introduction and indicate the hypotheses of the work. The end of the introduction is interesting, but it was a little confusing.
2 – The methods section failed to identify the methods that will be compared.
3 – The results are well written. I suggest explaining the p-values found.
4 – The discussion needs more depth. The authors express opinions about the method they think is best, but do not discuss the results found with other results in the literature. In this context, I suggest that: 1 - the discussion be redone; 2 – that the authors indicate whether the hypotheses (item 1) were confirmed or rejected; 3 – the results found are discussed, taking into account the p-value. In cases where the errors found were more significant than the gold procedure, take this result in comparison, 4 – Avoid recommending any method because the error was smaller. The results must be read faithfully, especially the significance values, and 5 – the authors must include their opinions on the selected methods in a practical application section.
5 – The conclusion must reflect the results found, considering the significance values. Descriptive results should be avoided.
Given the above, I affirm that the manuscript is interesting, but the authors must make mandatory corrections.
Sincerely,
Reviewer.
Author Response
1 – The introduction is well structured. However, I make the following suggestions:
1 – Remove the statement “In this paper we compare 27 methods of estimating vertical ground reaction forces (GRFs) from a single wearable accelerometer during running. GRFs are external reaction forces created with equal magnitude and opposite sense to the force that the foot applies to the ground with each step.”. It's meaningless. This is the purpose of the manuscript and should not appear in the first paragraph. Authors should review how to begin the introduction.
Thank you for the recommendation. The opening sentence has been removed.
2 – Explain the objective at the end of the introduction and indicate the hypotheses of the work. The end of the introduction is interesting, but it was a little confusing.
Thank you for pointing out this lack of clarity. The end of the introduction has been edited to improve clarity and better preface the structure and objectives of the article.
Given the goal of this paper is to compare methods to estimate parameters of the GRF there was no rationale for developing a priori hypotheses regarding which method was expected to best estimate any given GRF parameter under any given condition. Thus, no post hoc hypotheses have been included.
2 – The methods section failed to identify the methods that will be compared.
Thank you for pointing out this oversight. The methods to be compared are summarized in Table 2 which is now explicitly referenced in the Methods/Analysis section to facilitate readers finding the 27 methods being analyzed.
3 – The results are well written. I suggest explaining the p-values found.
An improved description of the p-values has now been included at the end of the methods section. No p-values are included for the main set of linear mixed effects models (LMEs) given all methods explain a significant amount of variation in the LMEs and we are not interested in whether any given participant, participant-method interaction, or participant-trial interaction explains a significant amount of model variance. Rather, the p-values included are for the second set of LMEs that are intended to examine the role of potential explanatory variables (running surface, speed, and foot strike angle). These p-values represent whether the model terms for running surface, speed, or foot strike angle explain a significant amount of variation in the model. In other words, whether model estimated forces are significantly affected by running surface, speed, or foot strike angle.
4 – The discussion needs more depth. The authors express opinions about the method they think is best, but do not discuss the results found with other results in the literature. In this context, I suggest that:
1 - the discussion be redone;
Changes have been made to the Discussion per the reviewers’ recommendations. Previous literature is explicitly compared in paragraph 4 of the Discussion. Methods replicated here performed as well or better than they performed in their original publication for 9 of 13 publications.
2 – that the authors indicate whether the hypotheses (item 1) were confirmed or rejected;
This is not hypothesis-driven research, rather it is a description of the currently available methods to estimate ground reaction force from wearable accelerometers. There was no rationale for the development of a priori hypotheses and no post hoc hypotheses have been added.
3 – the results found are discussed, taking into account the p-value. In cases where the errors found were more significant than the gold procedure, take this result in comparison,
Thank you for the recommendation. One sample t-tests with False Discovery Rate correction were conducted to compare each methods’ errors to the gold standard (0 error). The results tables have been updated to include this information.
4 – Avoid recommending any method because the error was smaller. The results must be read faithfully, especially the significance values, and
The current approach employs the best practices for method comparison studies (see: Bland & Altman, 1999; Myles, 2007; Carstensen, 2008; Carstensen, 2010; etc.) and we believe that our recommendations are a faithful interpretation of the data and results. To compare the errors across methods would require (m-1)(m)/2 multiple comparisons (with m representing the number of methods capable of estimating a given force feature; e.g., 300 comparisons for second peak alone). Given the volume of comparisons, interpreting the output would be extremely difficult and the extremely high Type I error rate would decrease confidence in any conclusions. Thus, rather than adding clarity to the paper, we believe that this approach would not be in line with best practices, would complicate interpretations, and would decrease confidence in the results. Further, RCs and LOAs have a single value (no distribution) for each method and feature and thus do not lend to comparison with p-values.
5 – the authors must include their opinions on the selected methods in a practical application section.
A “Practical Applications” section has been added to the Conclusions. This section includes the method we recommend applying, things to avoid/consider when applying (e.g., estimating first peak and loading rate), and a link to code that allows potential users to easily apply the methods.
5 – The conclusion must reflect the results found, considering the significance values. Descriptive results should be avoided.
Although we appreciate and share the reviewer’s concern with statistical rigor, this *is* a descriptive study that follows best practices for methods comparison studies (see: Bland & Altman, 1999; Myles, 2007; Carstensen, 2008; Carstensen, 2010; etc.) and, as detailed in our response to 4.4, we do not believe that comparisons across methods are appropriate here given the number of multiple comparisons that would be required. Thus, we believe that the conclusions accurately reflect the data and results.

Round 2
Reviewer 4 Report
Dear Editor,
I am grateful for the opportunity to evaluate, once again, the manuscript "Acceleration-Based Estimation of Vertical Ground Reaction Forces during Running: A Comparison of Methods Across Running Speeds, Surfaces, and Foot Strike Patterns". After reading, I observed that the authors improved the quality of the manuscript. However, I ask the authors to check the use of the acronym "ps". Throughout the text, the authors do not explain what this acronym is. There is no indication about this acronym in the statistical analysis, where it should be described. Without prior explanation, it appears that there was a typing error, and confused with the acronym "p" (p-value). After this small change, the manuscript can be approved.
Sincerely,
Reviewer.
Author Response
However, I ask the authors to check the use of the acronym "ps". Throughout the text, the authors do not explain what this acronym is. There is no indication about this acronym in the statistical analysis, where it should be described. Without prior explanation, it appears that there was a typing error, and confused with the acronym "p" (p-value). After this small change, the manuscript can be approved.
To improve clarity, all instances of "ps" in the manuscript have been replaced with "p values."